# A previously unrecognized membrane protein in the *Rhodobacter sphaeroides* LH1-RC photocomplex

Kazutoshi Tani [1,8✉], Kenji V. P. Nagashima[2], Ryo Kanno[3], Saki Kawamura[4], Riku Kikuchi[4], Malgorzata Hall[3], Long-Jiang Yu [5], Yukihiro Kimura [6], Michael T. Madigan[7], Akira Mizoguchi[1], Bruno M. Humbel[3] & Zheng-Yu Wang-Otomo [4,8✉]

*Rhodobacter* (*Rba.*) *sphaeroides* is the most widely used model organism in bacterial photosynthesis. The light-harvesting-reaction center (LH1-RC) core complex of this purple phototroph is characterized by the co-existence of monomeric and dimeric forms, the presence of the protein PufX, and approximately two carotenoids per LH1 αβ-polypeptides. Despite many efforts, structures of the *Rba. sphaeroides* LH1-RC have not been obtained at high resolutions. Here we report a cryo-EM structure of the monomeric LH1-RC from *Rba. sphaeroides* strain IL106 at 2.9 Å resolution. The LH1 complex forms a C-shaped structure composed of 14 αβ-polypeptides around the RC with a large ring opening. From the cryo-EM density map, a previously unrecognized integral membrane protein, referred to as protein-U, was identified. Protein-U has a U-shaped conformation near the LH1-ring opening and was annotated as a hypothetical protein in the *Rba. sphaeroides* genome. Deletion of protein-U resulted in a mutant strain that expressed a much-reduced amount of the dimeric LH1-RC, indicating an important role for protein-U in dimerization of the LH1-RC complex. PufX was located opposite protein-U on the LH1-ring opening, and both its position and conformation differed from that of previous reports of dimeric LH1-RC structures obtained at low-resolution. Twenty-six molecules of the carotenoid spheroidene arranged in two distinct configurations were resolved in the *Rba. sphaeroides* LH1 and were positioned within the complex to block its channels. Our findings offer an exciting new view of the core photocomplex of *Rba. sphaeroides* and the connections between structure and function in bacterial photocomplexes in general.

[1] Graduate School of Medicine, Mie University, Tsu 514-8507, Japan. [2] Research Institute for Integrated Science, Kanagawa University, 2946 Tsuchiya, Hiratsuka, Kanagawa 259-1293, Japan. [3] Imaging Section, Research Support Division, Okinawa Institute of Science and Technology Graduate University (OIST), 1919-1, Tancha, Onna-son, Kunigami-gun, Okinawa 904-0495, Japan. [4] Faculty of Science, Ibaraki University, Mito 310-8512, Japan. [5] Photosynthesis Research Center, Key Laboratory of Photobiology, Institute of Botany, Chinese Academy of Sciences, Beijing 100093, China. [6] Department of Agrobioscience, Graduate School of Agriculture, Kobe University, Nada, Kobe 657-8501, Japan. [7] School of Biological Sciences, Department of Microbiology, Southern Illinois University, Carbondale, IL 62901, USA. [8] These authors jointly supervised: Kazutoshi Tani, Zheng-Yu Wang-Otomo. ✉email: ktani@doc.medic.mie-u.ac.jp; wang@ml.ibaraki.ac.jp

*R*hodobacter (*Rba.*) *sphaeroides* is a purple phototrophic bacterium widely distributed in natural habitats. This organism is extraordinarily versatile in its abilities to grow by any of five metabolic modes: photoautotrophically ($CO_2$/light/anoxic); photoheterotrophically (organic compounds/light/anoxic); chemolithotrophically ($CO_2 + H_2$/dark/oxic); and by the respiration (dark/oxic) or fermentation (dark/anoxic) of organic compounds[1]. *Rba. sphaeroides* has been the most thoroughly investigated purple phototroph and has been widely used as a model for fundamental biochemical genetic studies of photochemistry, metabolism, and regulation. A crystal structure of the reaction center (RC) complex from *Rba. sphaeroides* was the second such to appear[2,3] following shortly after that of the RC from *Blastochloris* (*Blc.*) *viridis*[4]. The two structures displayed high similarities in the arrangements of most proteins and cofactors, suggesting that the core components of the RC—the L-subunit, M-subunit and cofactors—were structurally conserved among all purple bacteria[5].

The core light-harvesting (LH1) system of *Rba. sphaeroides* is unusual. The LH1 complex composed of α- and β-polypeptides forms an S-shaped structure in the native membranes[6,7] and 2D crystals[8–10]. The S-shaped LH1 incorporates two RCs resulting in a so-called dimeric LH1-RC complex, a structure that was also observed in *Rba. blasticus*[11] and *Rhodobaca bogoriensis*[12], and was suggested to exist in a few other purple bacteria[13]. A key component thought to be responsible for the dimerization is PufX, a protein present within the LH1-RC core complex[14,15]. PufX is an 82-amino-acid polypeptide encoded by the *pufX* gene in the *Rba. sphaeroides* photosynthetic gene cluster[16]. PufX is present in all species of *Rhodobacter*[17] and has been shown to be essential for phototrophic growth in both *Rba. sphaeroides* and *Rba. capsulatus*[18–20]. Previous work has probed the structural and functional role of PufX and implicated this key protein in regulation of membrane morphology, core photocomplex organization, and cyclic electron transfer[21].

In addition to the dimeric core complex, a monomeric LH1-RC-PufX complex also exists in native membranes of *Rba. sphaeroides* and *Rba. blasticus* and can be separately purified from the dimeric form[8,11,13]. The monomeric core complex is identical to one-half of a dimer with a gap in the LH1 ring, displaying a C-shaped arrangement[11]. It was demonstrated that monomers can reform dimers upon reconstitution, suggesting that the monomer represents a native state in equilibrium with the dimeric complexes[8]. Indeed, only the monomeric form has been observed for the core complexes from *Rba. veldkampii*[22–24] and *Rba. capsulatus*[13]. In addition to its structure, *Rba. sphaeroides* LH1 is also unique in its carotenoid amount. Regardless of dimeric or monomeric form, the *Rba. sphaeroides* LH1 complex contains approximately two carotenoids per αβ-polypeptides[25–28], the highest ratio among known LH1 complexes of purple phototrophs. However, despite the likely importance of carotenoids in LH1 function, *Rba. sphaeroides* LH1 carotenoids have not been resolved from low-resolution structural analyses (an 8-Å structure is the best thus far attained)[27]; thus, the precise position and function of carotenoids in the *Rba. sphaeroides* LH1 remains unknown.

Here we present a robust cryo-EM structure of the monomeric core complex from *Rba. sphaeroides* f. sp. *denitrificans* (strain IL106)[29]. A previously unrecognized integral membrane protein with a U-shaped conformation (designated as protein-U) was discovered in the LH1-RC structure and located to one side of the LH1 opening. Protein-U had been annotated as a hypothetical protein in the *Rba. sphaeroides* genome and is encoded in all *Rba. sphaeroides* strains with published genome sequences, implying a structural and functional role for this protein in the core complex. Our cryo-EM structure also reveals a precise conformation of PufX that definitively locates this crucial protein to a different

position in the complex than previously proposed. Finally, our structure allowed all carotenoids in the LH1 to be resolved and positioned in unique arrangements. Our findings shed new light on the key photosynthetic component of an old organism and provide a major new view of structure–function relationships in the core photocomplexes of photosynthetic bacteria.

## Results

**Structural overview.** The cryo-EM structure of the monomeric LH1-RC complex of *Rba. sphaeroides* IL106 was determined at 2.9 Å resolution (Fig. 1, Supplementary Table 1 and Supplementary Figs. 1–4). The LH1 complex is composed of 14 pairs of helical αβ-polypeptides, 28 BChls *a* and 26 spheroidenes surrounding the RC with a large opening in the slightly elliptical C-shaped ring structure. Although structures of the LH1 αβ-polypeptides and all proteins in the RC are consistent with previous work, it is notable that PufX was found at a different position and with a different conformation from that in the dimeric structure (PDB: 4JC9, 4JCB) determined at 8 Å for the *Rba. sphaeroides* LH1-RC (strain DBCΩG, Supplementary Fig. 5a)[27]. In addition, a large fragment of electron potential densities near the LH1-ring opening was observed in the refined map but could not be modeled by any known proteins or cofactors. Because our density map at 2.9 Å resolution was of very high quality, we were able to trace not only the main chain of this protein but also the sidechains for many residues. The derived amino acid sequence allowed us to search the *Rba. sphaeroides* IL106 genome (GenBank assembly accession: GCA_003363065.1), and in doing so, a hypothetical protein (DWF04_22265) containing 53 amino acids emerged whose sequence turned out to perfectly fit the density map (Fig. 2b). Hereafter, we refer to this previously unrecognized protein as "protein-U", the seventh distinct protein in the *Rba. sphaeroides* LH1-RC complex (Fig. 1c, d; Fig. 2a, b).

The *Rba. sphaeroides* LH1 αβ-polypeptides form a roughly elliptical C-shaped arrangement with long and short dimensions of 122 Å and 116 Å (Fig. 1b), respectively, for the outer β-polypeptide ring (distances measured between the outer edges of opposite helices). The LH1 α-polypeptides are formylated at their N-terminus as confirmed by TOF/MS (Supplementary Fig. 6) and the cryo-EM density map (Supplementary Fig. 4). These formyl groups were modeled in the LH1 structure and play a role in stabilizing the N-terminal regions of the α-polypeptides through hydrogen bonding. PufX (chain X) and protein-U (chain U) are located opposite each other on the two sides of the LH1-ring opening (Fig. 1, 2a). As a result, the positions of several LH1 αβ-subunits near protein-U gradually deviated from the closed LH1 ellipse (Fig. 1d) that has been observed in the monomeric type LH1-RCs from *Thermochromatium* (*Tch.*) *tepidum*[30] and *Rhodopseudomonas* (*Rps.*) *palustris*[31]. Molecules of BChl *a* in the *Rba. sphaeroides* LH1 are ligated by His residues with average His–Mg distances of 2.6 Å (α-His32) and 2.2 Å (β-His38), and have average Mg–Mg distances of 9.6 Å within a dimer and 8.4 Å between dimers (Fig. 1c, Supplementary Table 2). Cofactors in the RC include four BChls *a*, two bacteriopheophytins (BPhe) *a*, and one 15-*cis*-spheroidene and ubiquinone (UQ)-10 molecules at both the $Q_A$ and $Q_B$ site. Non-heme iron and the head group of UQ in the RC are aligned along a line parallel to that connecting PufX and protein-U. A total of three UQ-10, four cardiolipins, and 13 phosphatidylglycerols were modeled in the LH1-RC structure of *Rba. sphaeroides* IL106. The numbers of phospholipids are in agreement with those determined from biochemical analysis[32], and one of the cardiolipins was found at the same position as that reported previously in an RC-only structure from *Rba. sphaeroides*[33].

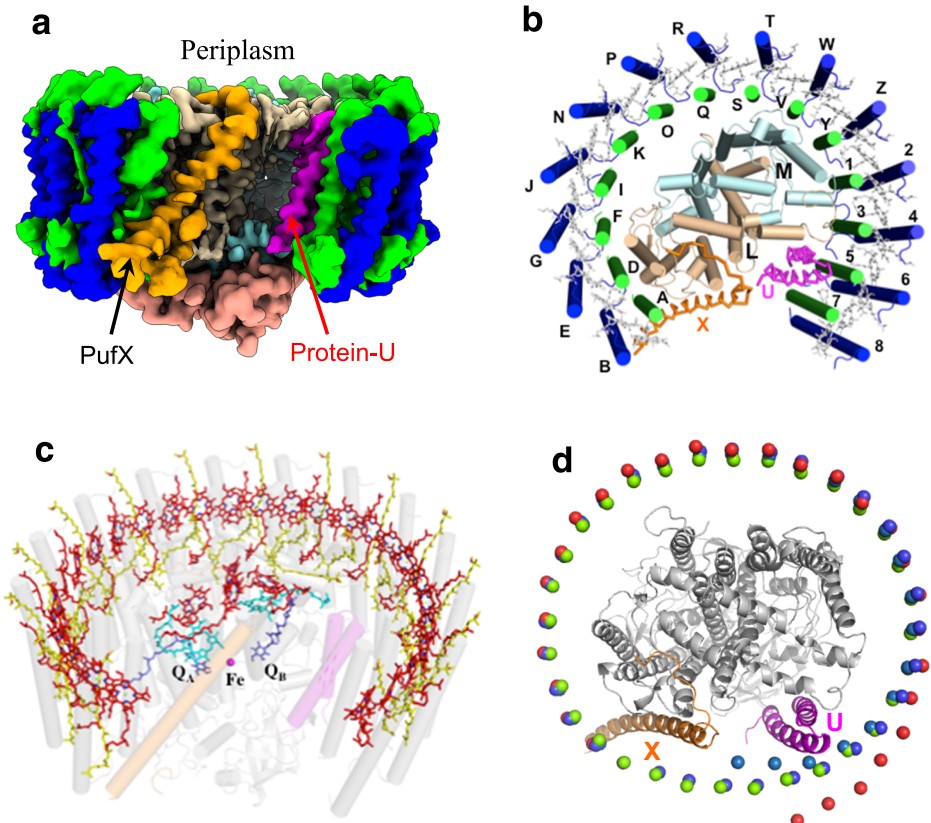

**Fig. 1 Structure overview of the monomeric LH1-RC complex from *Rba. sphaeroides* IL106. a** Side view of surface representations for the LH1-RC proteins parallel to the membrane plane. **b** Top view of the LH1-RC proteins from the periplasmic side of the membrane. Labels indicate chain IDs, and the RC H-subunit was omitted for clarity. Color scheme: LH1-α, green; LH1-β, blue; RC-L, wheat; RC-M, cyan; RC-H, salmon; PufX, orange; protein-U, magenta. **c** Tilted view of the cofactor arrangement viewing from the periplasmic side. Color scheme: BChl *a*, red sticks; spheroidenes, yellow sticks; BPhe *a*, cyan sticks; UQ-10, blue sticks; PufX, transparent orange cylinder; protein-U, transparent magenta cylinder, and transparent gray cylinders for all other proteins. **d** Comparisons of the central Mg atom positions in the LH1 BChl *a* molecules from *Rba. sphaeroides* IL106 (red spheres) with those from *Tch. tepidum* (green spheres, PDB: 5Y5S), protein-W-containing *Rps. palustris* (cyan spheres, PDB: 6Z5S), and protein-W-deficient *Rps. palustris* (blue spheres, PDB: 6Z5R). The structures were superimposed by Cα carbons of the RC (L, M, H) proteins. The ribbon models represent the RC-L, -M and -H subunits (gray), PufX (orange), and protein-U (magenta) of the *Rba. sphaeroides* IL106.

**The unidentified protein-U.** *Rba. sphaeroides* protein-U is located in the interior of the LH1-RC complex between the RC-L subunit and two LH1 α-polypeptides and exhibits a helix-turn-helix conformation with both N- and C-termini on the cytoplasmic side (Fig. 2a, b). The two transmembrane (TM) helical regions contain highly hydrophobic residues rich in Gly and Ala, and are connected by a short loop of five residues. The characteristic U-shaped feature was clearly observed in the cryo-EM map and is consistent with that predicted by the membrane protein topology program TMHMM[34] (Fig. 2d). Protein-U interacts with surrounding proteins mainly through its terminal domains and loop region. The main chain oxygen of protein-U Val4 and the sidechain carboxyl group of Glu6 form hydrogen bonds with the sidechain of Arg15 in an α-polypeptide, while a segment (Thr49–Pro50–Asn51) in the protein-U C-terminal region forms hydrogen bonds with two α-polypeptides (chains 5 and 7) (Fig. 1b, 2c). In the loop region of protein-U, Trp32 and Phe33 form close contacts with Trp265 and Trp266 in the RC-L subunit where a putative UQ-10 molecule is present nearby. The unique position and interaction pattern of protein-U indicate that one of its functions may be as a "spacer" pushing several nearby LH1 polypeptides outward from the RC as shown in Fig. 1d and implying a structural role in producing steric hindrances to prevent the formation of a closed LH1 ring.

To further investigate potential functions of protein-U, we constructed a mutant strain (IL106-ΔU) deficient in protein-U (Supplementary Fig. 7). This deletion mutant was capable of phototrophic growth at a growth rate similar to that of wild-type cells. However, levels of dimeric LH1-RC significantly decreased in the mutant compared with the wild type. Sucrose density gradient analysis revealed that the ratio of dimeric to monomeric LH1-RC was cut by nearly two-thirds (0.44 to 0.15) in the mutant (Fig. 2e, Supplementary Fig. 8a), indicating that protein-U likely plays structural and functional roles in the stabilization of the monomer and formation of the dimers of the LH1-RC complex. No apparent effects of the absence of protein-U on absorption spectra of either dimeric or monomeric LH1-RCs were detected (Supplementary Fig. 8b).

**The PufX polypeptide.** The full-length PufX of *Rba. sphaeroides* IL106 is encoded by 82 amino acids in which the N-terminal Met and C-terminal 13 residues are post-translationally removed in the expressed polypeptide as confirmed by TOF/MS (Fig. 3d, Supplementary Fig. 6). The PufX sequence of strain IL106 differs slightly from that of the type strain 2.4.1 at Asp58 and Pro61, where both are replaced with Gln in *Rba. sphaeroides* strain 2.4.1. In our cryo-EM structure, the N-terminal 15 residues of the expressed PufX were invisible presumably due to disordered conformations. PufX is located on the opposite side of protein-U

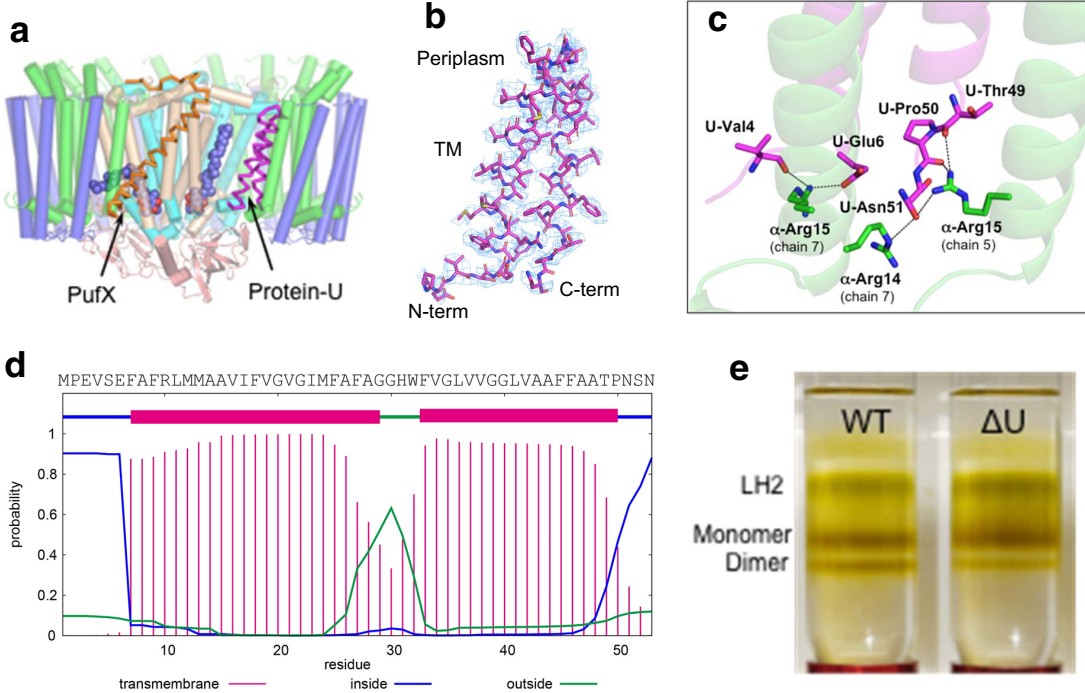

**Fig. 2 Structure and property of *Rba. sphaeroides* protein-U. a** Relative position of protein-U (magenta ribbon) in the LH1-RC. PufX is shown by the orange ribbon and the UQ-10 at $Q_A$ and $Q_B$ sites are shown by blue spheres. All other proteins are shown by cylinders with the same color scheme as in Fig. 1(a). **b** Structure of protein-U and its density map at a contour level of 3.0σ. **c** Close contacts (<3.5 Å, dashed lines) between the protein-U terminal regions (magenta) and LH1 α-polypeptides (green). **d** Amino acid sequence of protein-U and its two consecutive transmembrane regions predicted by the membrane protein topology program TMHMM. **e** Sucrose density gradient (10–40% w/v) centrifugations of the solubilized pigment-protein complexes from wild-type (WT) and protein-U-deleted (ΔU) *Rba. sphaeroides* IL106 membranes.

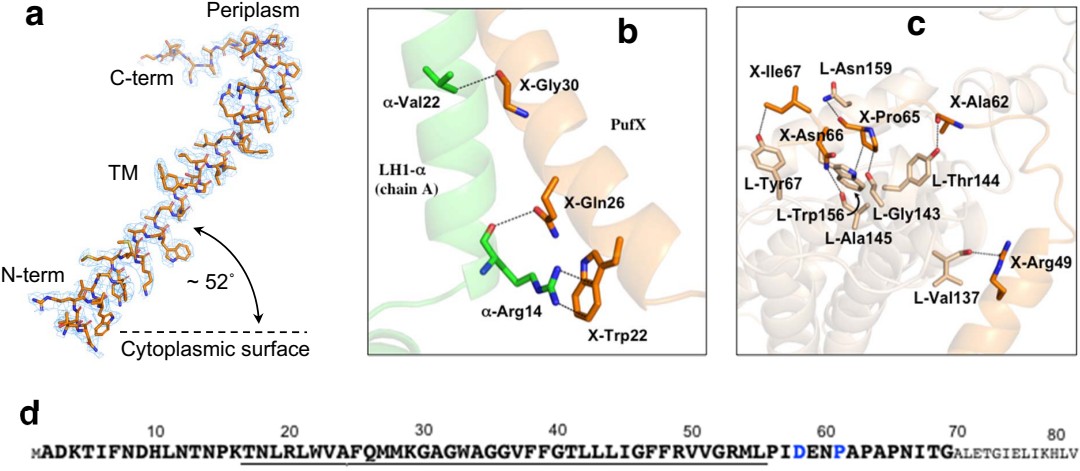

**Fig. 3 Structure and interactions of *Rba. sphaeroides* PufX. a** Structure and density map (3.0σ) of PufX with orientation toward the presumed membrane plane. **b** Close contacts (<3.5 Å, dashed lines) between the PufX N-terminal region (transparent orange) and an LH1 α-polypeptide (chain ID: A, transparent green). **c** Close contacts (<3.5 Å, dashed lines) between the PufX C-terminal domain (transparent orange) and the RC L-subunit (transparent wheat). **d** Primary sequence of the full-length PufX in *Rba. sphaeroides* IL106. Sequence for the expressed PufX is shown by larger bold fonts. Membrane-spanning residues are underlined. The Glu58 and Pro61 (blue fonts) are both replaced by Gln in the type strain 2.4.1.

in the LH1-ring opening (Fig. 2a) at the same position with almost the same conformation as that of another PufX-containing LH1-RC recently reported for *Rba. veldkampii* (Supplementary Fig. 5b)[24]. PufX has a single, unusually long transmembrane domain with its N-terminus on the cytoplasmic side and C-terminus on the periplasmic surface (Fig. 3a). Differing from other proteins in the LH1-RC, PufX displays a remarkably tilted conformation for its transmembrane helical domain with an angle of approximately 52° inclined toward the presumed membrane

plane (Fig. 3a); this results in a much longer membrane-spanning region (38 residues, Supplementary Fig. 10c) in good agreement with an experimental estimation for the core segment in PufX[35].

No direct interactions (<4.0 Å) were observed between PufX and pigment molecules (BChl *a* and carotenoids). However, the N-terminal region of PufX interacts with an LH1 α-polypeptide (chain A), while the C-terminal domain has contacts with the RC L-subunit on the periplasmic surface. The Trp22 in PufX likely has cation–π interactions with the sidechain of Arg14 in an

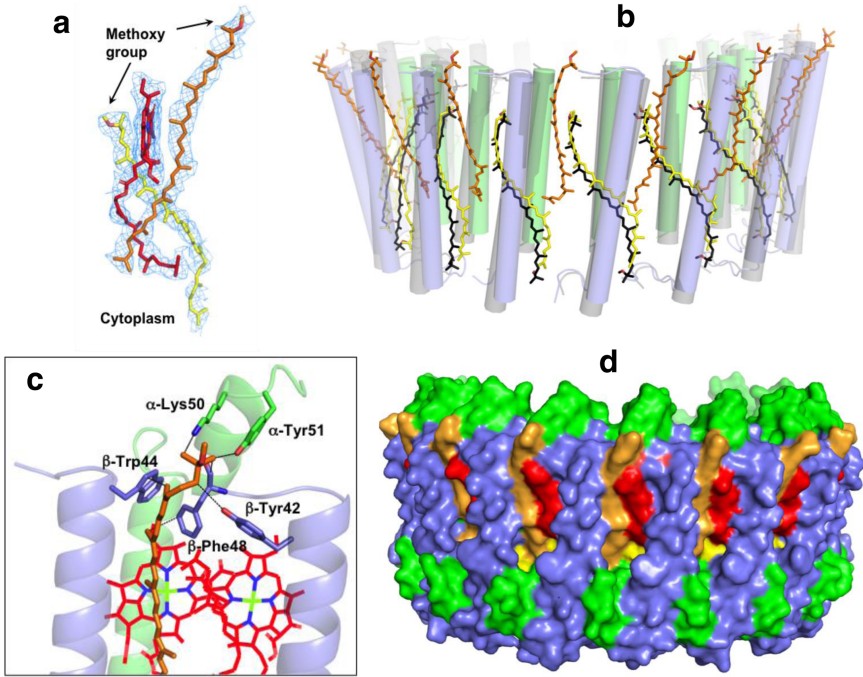

**Fig. 4 Carotenoids in the *Rba. sphaeroides* LH1. a** Typical structures and density maps (3σ) of two spheroidenes (yellow and orange) relative to the position of a BChl *a* (red) in the LH1 complex with different configurations. **b** Two groups of the spheroidenes (yellow and orange sticks for Group-A and Group-B, respectively) in the *Rba. sphaeroides* LH1 in comparison with the carotenoids in the *Tch. tepidum* LH1 (black sticks, PDB:5Y5S). The structures were superimposed by Cα carbons of the LH1 αβ-polypeptides between *Rba. sphaeroides* (transparent green and slate-blue cylinders) and *Tch. tepidum* (transparent gray cylinders). **c** Close contacts (<4.0 Å, dashed lines) between a Group-B spheroidene and its surrounding αβ-polypeptides. **d** Side view of the surface representation shows a sealed fence for the *Rba. sphaeroides* LH1 complex. Color scheme: LH1-α, green; LH1-β, slate-blue; BChl *a*, red; spheroidenes, yellow (Group-A) and orange (Group-B).

α-polypeptide (Fig. 3b). The Arg49 and Ala62 in PufX form hydrogen bonds with Val137 and Thr144 in the RC L-subunit, respectively (Fig. 3c). The extensive interactions between the PufX C-terminal end (Ala62–Ile67) and a segment of the RC L-subunit (Gly143–Trp156) explain the experimental observation that the C-terminus of PufX plays an important role in dimerization and assembly of the LH1-RC in *Rba. sphaeroides*[36].

**Carotenoids in the *Rba. sphaeroides* LH1 Complex**. A total of 26 all-*trans*-spheroidenes in the LH1 complex of *Rba. sphaeroides* IL106 were clearly resolved in our cryo-EM map (Fig. 1c, 4a, Supplementary Fig. 4). This corresponds to a ratio of approximately two carotenoids per αβ-polypeptides in contrast to the one carotenoid per αβ-pair commonly found in other LH1 complexes[24,30,31,37–39]. All methoxy groups of the *Rba. sphaeroides* carotenoids point toward the periplasmic side, and the carotenoids can be grouped into two sets based on their orientations and positions in the LH1. Group-A spheroidenes are embedded deeply in the transmembrane region between the α- and β-polypeptides at similar positions and with a similar conformation to those in the LH1 complexes of other purple bacteria[24,30,31,37–40]. A representative comparison is shown in Fig. 4b for the conformations between the carotenoids in *Rba. sphaeroides* IL106 and that of *Tch. tepidum*. The methoxy groups of these carotenoids are positioned close to the center of bacteriochlorins (Fig. 4a) and there are no apparent associations between the carotenoids and surrounding polypeptides and BChl *a* molecules.

In contrast to Group-A spheroidenes, Group-B spheroidenes adopt a distinct configuration within the *Rba. sphaeroides* LH1 with a large shift toward the periplasmic side (Fig. 4b). Each of these carotenoids is located in between two adjacent αβ-pairs with

their methoxy group protruded on the periplasmic surface. The methoxy ends of these carotenoids are in close proximity to the C-terminal ends of αβ-polypeptides (Fig. 4c) and the central portions form hydrophobic interactions with BChl *a* molecules. The Group-B spheroidenes, combined with Group-A spheroidenes, fill the space between adjacent αβ-subunits (Fig. 4d), and this leads to blockage of the pores (channels) observed in BChl *a*-containing LH1 complexes containing only Group-A carotenoids[30,31,38,39]. The combination of both Group A and B carotenoids results in a tightly sealed, impenetrable LH1 "fence" that requires alternative strategies for facilitating quinone transport between the *Rba. sphaeroides* RC and quinone pools during photosynthetic electron flow. These alternative strategies include the C-shaped monomeric LH1 whose structure was revealed herein or an S-shaped dimeric LH1 with incomplete rings as previously reported[27].

## Discussion

The gene encoding the 53 amino acids in protein-U is flanked in the *Rba. sphaeroides* IL106 genome by genes encoding a transglutaminase family protein and metallo-hydrolase and is the only gene in this region transcribed in the opposite direction (Supplementary Fig. 7). This suggests that transcription of the gene encoding protein-U is controlled independently from that of adjacent genes. The expressed protein-U locates in the interior space between LH1 (chains 5 and 7) and RC near the edge of the LH1-ring opening. The corresponding position is occupied by a cardiolipin molecule in the protein W-containing *Rps. palustris* LH1-RC that also has an opening in the LH1 ring[31]. Protein-U has a tandem arrangement of two membrane-spanning segments connected by a short loop, forming a U-shaped conformation. High frequencies of Gly and Ala residues in the transmembrane

region indicate a tendency for these α helices to pack tightly at particular sites such as Gly22, Gly35, and Gly39 (Fig. 2d) to form a canonical left-handed supercoil as is often seen in α-bundle membrane proteins[41–43]. Such helical pairs with antiparallel arrangements are known to form large interfacial areas, which in the case of protein-U may contribute to LH1-RC stabilization and/or facilitate dimerization of the complex.

A genomic database search of the genus *Rhodobacter* revealed that six of fifteen species contain protein-U (or protein-U-like) genes, and all *Rba. sphaeroides* strains with sequenced genomes encode this protein (Supplementary Fig. 9a). Proteins-U can be classified into three types based on their sequences (Supplementary Fig. 9b): Type-1 (WP_002721225) for *Rba. johrii, Rba. megalophilus* and all but one *Rba. sphaeroides* strains; Type-2 (WP_176504535) for *Rba. ovatus*; and Type-3 (WP_085996593) for *Rba. azotoformans, Rba. sediminicola* and *Rba. sphaeroides* ATCC 17025. Type-1 and Type-3 proteins-U have the same number of residues and share high sequence identity, whereas Type-2 has a long N-terminal domain but also shares high sequence identity with Type-1 and Type-3 proteins-U in other portions (Supplementary Fig. 9b). All *Rhodobacter* species containing protein-U also have *pufX* (or *pufX*-like) genes but the converse is not true. Considering that a *Rba. sphaeroides* protein-U-deletion strain in our work and wild-type *Rba. blasticus* that naturally lacks protein-U[11] are both capable of photosynthetic growth and producing at least some dimeric LH1-RC, protein-U is clearly not indispensable for photosynthesis as is true of the more widely distributed PufX; a PufX-deletion strain of *Rba. sphaeroides* (ATCC 17023) was reported to be photosynthetically incompetent[19,44]. Therefore, we conclude that major functions of protein-U should include (i) stabilization of the partially opened LH1-RC structures and (ii) enhancement of efficiency for dimerization of the LH1-RC complex from that formed spontaneously, suggesting that the dimeric form may be the most efficient (or preferred) state of the complex in *Rba. sphaeroides*.

The small size and extremely high hydrophobicity of *Rba. sphaeroides* protein-U likely hindered its early discovery by biochemical and structural analyses. The presence of an additional protein in dimers of the *Rba. sphaeroides* LH1-RC was previously suspected from extra residual densities in an 8.5-Å cryo-EM projection map of 2D crystals from *Rba. sphaeroides* strain DD13/DGa2[10]. Two significant density features were identified within each half of the LH1-RC dimer, corresponding approximately to the positions of protein-U and PufX in our cryo-EM structure of the monomeric LH1-RC. One of the densities was tentatively proposed to be PufX[10] but this corresponds to protein-U in our structure, while another relatively diffuse density feature was left unassigned[10] and this corresponds to PufX in our study. Subsequently, a 3D structure of the dimeric LH1-RC from *Rba. sphaeroides* strain DBCΩG was determined at 8 Å using combined techniques[27]. However, PufX was found at yet a different position in this structure from those previously reported[10] and in our work (Supplementary Fig. 5a). The two *Rba. sphaeroides* strains used in these studies both lacked a peripheral light-harvesting complex (LH2) but were mutant derivatives of the same parental strain (wild-type: NCIB8253)[45]. Thus, the conflicting results likely indicate the difficulty in obtaining a definitive structure of such a large photocomplex at limiting resolutions, although structural rearrangements induced by genetic manipulations and/or dimerization cannot be absolutely ruled out.

Structures for individual polypeptides of LH1-β and PufX isolated from *Rba. sphaeroides* have been investigated using solution NMR. The β-polypeptide shows a single transmembrane α-helix starting from Leu19 with a remarkably high degree of disorder for the N-terminal domain in organic solvents[46]. The

flexible and unstructured features for the N-terminal region of the β-polypeptides were also observed in our cryo-EM structure in which the first 5–10 residues were invisible in the density map. Subsequently, the solution structures of the β-polypeptide were further determined in organic and detergent solvents, both exhibiting two helical domains separated by a flexible region with largely bent conformations[47,48] in contrast to that observed in our cryo-EM structure. Solution structures of PufX were also determined in organic solvents, yielding two quite different conformations: one showed two hydrophobic helices connected by a helical bend (PDB: 2ITA, 2NRG; Supplementary Fig. 10b)[49,50], whereas a second revealed a rather straight, single continuous helix for the central transmembrane domain (PDB: 2DW3)[51]. The latter closely matches that in our cryo-EM structure over the α-helical region (Supplementary Fig. 10a).

Alignments of eighteen PufX and PufX-like sequences in the database revealed a well-conserved central helical domain flanked by flexible termini (Supplementary Fig. 10c), and the diversities in structure, function and interactions of PufX have been thoroughly reviewed[21]. Genetic studies have demonstrated that *Rba. sphaeroides* PufX is most tightly associated with the LH1-RC complex, less tightly bound to the LH1-only complex, and least tightly associated with RC-only preparations[52]. PufX tends to co-purify with the LH1 α-polypeptide[53], implying preferential interactions with the LH1 inner ring (chain A), consistent with our cryo-EM structure. Cross-species studies of PufX led to a hypothesis that in all *Rhodobacter* species PufX defines the composition of the monomeric LH1-RC complex, managing the aggregation state of the pigment proteins around the RC and determining the position of the LH1-ring opening relative to the entrance of the $Q_B$ pocket in the RC[13]. The nearly symmetric arrangements of PufX and protein-U relative to the LH1-ring opening and the fact that deletion of protein-U resulted in a greatly reduced amount of dimeric LH1-RC may suggest a similar and/or cooperative role for protein-U in controlling the topology of the LH1 complex.

PufX in *Rba. sphaeroides* likely correlates with the carotenoid content in LH1. It was demonstrated that an engineered *Rba. sphaeroides* LH1 with a point-mutation (LH1-α Trp8→Phe) produced fewer carotenoids and allowed photosynthetic growth in the absence of PufX[54]. This result was interpreted by assuming that an LH1 with fewer carotenoids would possess multiple pores (channels) in the closed complex as seen in the LH1 of *Tch. tepidum*[30] and that this would allow quinone transport through the pores in the absence of PufX and support photosynthetic growth. By contrast, a corresponding *Rba. sphaeroides* strain with a higher carotenoid content in its LH1 was photosynthetically incompetent[54]. This in turn suggested that the high carotenoid content (approximately two carotenoids per αβ-pair) in the native *Rba. sphaeroides* LH1 could block the LH1 channels (as confirmed in our structure, Fig. 4d) requiring alternative pathways for quinone transport in order to grow photosynthetically. Consequently, a PufX-containing C-shaped (monomeric) or S-shaped (dimeric) LH1 complex could fulfill this physiological requirement.

As experimental model systems, phototrophic purple bacteria such as *Rba. sphaeroides* continue to reveal the most intimate secrets of photosynthetic energy conversion, and in this regard, our work provides the most robust and detailed structure of the *Rba. sphaeroides* LH1-RC structure to date. Our cryo-EM structure not only identified a previously unsuspected integral membrane protein and characterized its likely functions but also determined the precise position and conformation of PufX—a long-standing issue in bacterial photosynthesis—and resolved all carotenoids in the core complex. Our findings lay the foundation for an even deeper understanding of photosynthesis, in particular

those aspects surrounding structure–function relationships in light-energy conversion, quinone transport, and regulation of carotenoid biosynthesis.

## Methods

**Preparation and characterization of the native LH1-RC complex.** The *Rba. sphaeroides* f. sp. *denitrificans* (strain IL106) cells were cultivated phototrophically (anoxic/light) at room temperature for 7 days under incandescent light (60 W). This strain can also grow chemoheterotrophically using nitrate and dimethyl sulfoxide (DMSO) as the terminal electron acceptors (dark/anoxic)[29]. Preparation of the native LH1-RC was conducted by solubilizing chromatophores ($OD_{870-nm} = 40$) with 1.0 % w/v *n*-dodecyl-β-D-maltopyranoside (DDM) in 20 mM Tris-HCl (pH 8.0) buffer for 60 min at room temperature, followed by differential centrifugation. The supernatant was loaded onto a DEAE column (Toyopearl 650 S, TOSOH) equilibrated at 4 °C with 20 mM Tris-HCl buffer (pH 8.0) containing 0.1 % w/v DDM. The fractions were eluted in an order of LH2, monomeric LH1-RC, and dimeric LH1-RC by a linear gradient of NaCl from 0 mM to 400 mM. The peak fractions of monomeric LH1-RC were collected and further purified by sucrose gradient density centrifugation with five-stepwise sucrose concentrations (10, 17.5, 25, 32.5, and 40% w/v) in 20 mM Tris-HCl buffer (pH 8.0) containing 0.05% w/v DDM at 4 °C and 150,000 × g for 6 h. The monomeric LH1-RC fractions were concentrated for absorption measurement and assessed by negative-stain EM using a JEM-1011 instrument (JEOL) (Supplementary Fig. 1a). We were unable to distinguish the native monomeric LH1-RC originally present in the intact membranes from those degraded from dimers during isolation and purification. Masses and composition of the LH1 and PufX polypeptides were measured by matrix-assisted laser desorption/ionization time-of-flight mass spectroscopy (MALDI-TOF/MS) and reverse-phase HPLC (Supplementary Fig. 6b, 6c), respectively[55]. Quinones were extracted from chromatophores, purified LH1-RC, and RC-only complexes, and quinone contents were analyzed (Supplementary Fig. 6d)[56]. Approximately six UQ-10 molecules were estimated per purified LH1-RC.

**Mutagenesis and characterization.** A 0.8-kb DNA region at the direct upstream and a 0.9-kb DNA region at the direct downstream of the protein-U gene were amplified by PCR using *Rba. sphaeroides* IL106 genomic DNA as the template and the PCR primers shown in Supplementary Table 3. These DNA fragments were connected in a manner that excludes the protein-U gene and cloned in a suicide vector pJPCm[57] using an In-Fusion HD Cloning Kit (TAKARA BIO, Shiga, Japan) as shown in Supplementary Fig. 7. This plasmid was named pJPCm_ΔU, and then a DNA fragment containing *sacRB* genes (lethal genes under the presence of sucrose) and a kanamycin-resistant cartridge were inserted at the unique *Sac*I restriction site on this plasmid. This plasmid was named pJPCm_ΔU-SKm and introduced from the *E. coli* S17-1 λpir host cells by conjugational transfer into the cells of *Rba. sphaeroides* IL106 wild-type strain with a spontaneous resistance to rifampicin. The gene encoding protein-U was deleted from the *Rba. sphaeroides* genome *via* a two-step homologous recombination, which was screened by kanamycin resistance in the first step and by sucrose resistance in the second, as previously described[57]. After the conjugation *E. coli* cells were eliminated by addition of 50 μg/ml of rifampicin to the growth medium. The removal of the protein-U gene without any insertions of antibiotics-resistant cartridges was confirmed by PCR and DNA sequencing experiments. This mutant was named IL106-ΔU.

Cells of *Rba. sphaeroides* IL106-ΔU were cultivated phototrophically at room temperature under the same condition as that for the wild-type. The chromatophores of both wild-type and IL106-ΔU were treated with 1.0% w/v DDM in 20 mM Tris-HCl (pH 8.0) buffer for 60 min at room temperature, followed by differential centrifugation. The supernatants (1 mL) with adjusted concentrations of $OD_{870-nm} = 5$, 9, 14 were loaded on a stepwise sucrose gradient (10, 17.5, 25, 32.5, and 40% w/v) in 20 mM Tris-HCl buffer (pH 8.0) containing 0.05 % w/v DDM. Centrifugation was conducted at 4 °C and 150,000 × g for 6 h. All of the dimeric and monomeric LH1-RC layers were carefully collected and a ratio of the dimeric to monomeric LH1-RC was calculated using their volumes and absorption intensities (Supplementary Fig. 8).

**Cryo-EM data collection.** Proteins for cryo-EM were concentrated to ~3 mg/ml. Two microliters of the protein solution were applied on glow-discharged holey carbon grids (200 mesh Quantifoil R2/2 molybdenum), which had been treated with $H_2$ and $O_2$ mixtures in a Solarus plasma cleaner (Gatan, Pleasanton, USA) for 30 s and then blotted, and plunged into liquid ethane at –182 °C using an EM GP2 plunger (Leica, Microsystems, Vienna, Austria). The applied parameters were a blotting time of 6 s at 80% humidity and 4 °C. Data were collected on a Talos Arctica (Thermo Fisher Scientific, Hillsboro, USA) electron microscope at 200 kV equipped with a Falcon 3 camera (Thermo Fisher Scientific) (Supplementary Fig. 1). Movies were recorded using EPU software (Thermo Fisher Scientific) at a nominal magnification of 92 k in counting mode and a pixel size of 1.094 Å at the specimen level with a dose rate of 0.98 e- per physical pixel per second, corresponding to 0.82 e- per $Å^2$ per second at the specimen level. The exposure time was 51 s, resulting in an accumulated dose of 42 e- per $Å^2$. Each movie includes 40 fractioned frames.

**Image processing.** All of the stacked frames were subjected to motion correction with MotionCor2[58]. Defocus was estimated using CTFFIND4[59]. A total of 551,846 particles were selected from 2,766 micrographs using the EMAN2 suite (Supplementary Fig. 2)[60]. The initial 3-D model was generated with 36,284 particles from 87 selected micrographs with underfocus values ranging between 2 and 3 μm using EMAN2. All of the picked particles were further analyzed with RELION3.0 and 3.1[61], and 219,927 particles were selected by 2D classification and divided into four classes by 3D classification resulting in only one good class containing 160,488 particles. The 3D auto refinement without any imposed symmetry (C1) produced a map at 3.10 Å resolution after contrast transfer function refinement, Bayesian polishing, masking, and post-processing. The selected 160,488 particle projections were subjected to subtraction of the detergent micelle density followed by 3D auto refinement to yield the final map with a resolution of 2.94 Å according to the gold-standard Fourier shell correlation using a criterion of 0.143 (Supplementary Fig. 2)[62]. The local resolution maps were calculated on RESMAP[63].

**Model building and refinement of the LH1-RC complex.** The atomic models of the LH1 ring of the *Tch. tepidum* LH1-RC (PDB: 5Y5S) and the RC of the *Rba. sphaeroides* (PDB: 1PCR) was fitted to the cryo-EM map obtained for the *Rba. sphaeroides* LH1-RC using Chimera[64]. Amino acid substitutions and real-space refinement for the peptides and cofactors were performed using COOT[65]. Whole regions of PufX and protein-U as well as both terminal regions of the LH1 α-subunit were modeled ab-initio based on the density. The manually modified model was refined in real-space on PHENIX[66], and the COOT/PHENIX refinement was iterated until the refinements converged. Finally, the statistics calculated using MolProbity[67] were checked. Figures were drawn with the Pymol Molecular Graphic System (Schrödinger)[68] and UCSF Chimera[64].

**Reporting Summary.** Further information on research design is available in the Nature Research Reporting Summary linked to this article.

## Data availability

Map and model have been deposited in the EMDB and PDB with the accession codes: EMD-31400 and 7F0L. The coordinates (5Y5S) for *Tch. tepidum* LH1-RC is available at PDB. All other data are available from the corresponding authors upon reasonable request.

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

## Acknowledgements

We thank Megumi Kobayashi, Ayumi Imai, and Risa Onuma for providing excellent technical assistances. This research was partially supported by Platform Project for Supporting Drug Discovery and Life Science Research (Basis for Supporting Innovative Drug Discovery and Life Science Research (BINDS)) from AMED under Grant Numbers JP20am0101118 (support number 1758) and JP20am0101116 (support number 1878), 17am0101116j0001, 18am0101116j0002, and 19am0101116j0003. R.K., M.H., and B.M.H. acknowledge the generous support of the Okinawa Institute of Science and Technology and the Japanese Cabinet Office. L.-J.Y. acknowledges support of the National Key R&D Program of China (No. 2019YFA0904600). This work was supported in part by JSPS KAKENHI Grant Numbers JP16H04174, JP18H05153, JP20H05086, and JP20H02856, Takeda Science Foundation, and the Kurata Memorial Hitachi Science and Technology Foundation, Japan.

## Author contributions

Z.-Y.W.-O. and K.T. designed the work, K.T., K.V.P.N., R. K., S.K., R. K., and M.H. performed the experiments, K.T., K.V.P.N., R. K., L.-J.Y., Y.K., M.T.M., A.M., B.M.H., and Z.-Y.W.-O. analyzed data, Z.-Y.W.-O., K.T., K.V.P.N., and M.T.M. wrote the manuscript. All authors edited and revised the manuscript.

## Competing interests

The authors declare no competing interests.
