## [Peer Review File · Nature Communications]

A Previously Unrecognized Membrane Protein in the Rhodobacter sphaeroides LH1-RC PhotocomplexREVIEWER COMMENTS

Reviewer #1 (Remarks to the Author):

This is a sound piece of structural biology carried out on a monomeric RC-LH1 complex from *Rb. sphaeroides*. The novel features are a better resolution, the presence of a new component 'U' that seems to control the amount of dimer form made, the first visualisation of the two carotenoid molecules per alpha/beta LH1 apoprotein pair and the idea that the second Car stops the possibility of quinone transfer through the LH1 apoprotein barrier. The paper really misses a trick though. It would have been much more significant if a structure had also been determined for the monomer lacking the 'U' component and for the dimer form. Then the really exciting structural role of 'U' and what controls dimer formation could have been seen.

Minor comments;

This is not near atomic resolution as stated in the introduction.

The first ref showing the Car:Bchl_a ratio in the type of complex was Sistrom W.R. In 'The Photosynthetic Bacteria' pgs. 841-848 (Eds. Clayton and Sistrom) 1978

Although spheroidene is the major carotenoid when the cells are harvested, membranes are made and the complex is prepared the presence of oxygen results in the conversion of some of the spheroidene into spheroidenone. Did the authors determine the ratio of these two carotenoids to be sure the assumption of just using spheroidene in the structure is correct.

At this resolution it is hard to be sure which type of phospholipid is present. It would be better to reference that the RC has been shown to bind cardiolipin to strengthen this assignment, viz. McAukey et al. (1999) PNAS 96, 14706-14711.

Most readers will not know that the methoxy group at the end of spheroidene is at one end. This should be made much clearer in the text or properly label in a fig.

Reviewer #2 (Remarks to the Author):

Photosynthesis is the global process by which photosynthetic organisms transform light energy into chemical energy. Photosynthesis is a primary source for the organic synthesis from inorganic matter. Within various photosynthetic organisms, the purple bacteria place the important niche, as prokaryotes, which perform photosynthetic reactions in their way. This feature is one of the reasons why purple bacteria are metabolically the most versatile organisms on Earth.

The interest in the studies of photosynthesis of purple bacteria led to the Nobel Prize in Chemistry 1988 award for Prof. Dr Hartmut Michel, Prof. Dr Johann Deisenhofer and Prof. Dr Robert Huber.

The award was given for the study of the three-dimensional molecular structure and the mode of action of the reaction centre of photosynthesis in the purple bacterium *Rhodospseudomonas viridis*. Ever since the number of research was constantly growing, revealing many new species of purple bacteria and novel organisation of pigment-protein complexes taking part in photosynthetic reactions.

Here I reviewed the manuscript named: "A Novel Membrane Protein in the *Rhodobacter sphaeroides* LH1-RC Photocomplex."

This is a collaborative work performed by an international group of scientists.

The main novelties that are illuminated in the manuscript are:

1) For the first time, the monomeric light-harvesting reaction centre core complex (RC-LH1-PufX) from purple bacteria *Rb. sphaeroides* strain IL106 is revealed at such a high resolution (2.9 Å).

Due to such resolution, all the protein secondary structures and many side-chains, pigments and cofactors were undoubtedly defined, modelled and located inside the cryo-EM map (based on the figures//and supplementary figures).

2) A high amount of carotenoids is revealed: each LH1 antenna subunit contains two carotenoids per one $\alpha\beta$ -polypeptides.

This ratio is the highest of any currently known in the reactions centre of purple bacteria.

3) The structure of unknown density, so-called "protein-U", was found.

Previous attempts to characterise this density were unsuccessful due to its biochemical properties or low resolution of the 3D map. Previous studies defined this density as belonging to the PufX protein.

4) The authors proposed the following explanation of the functioning of "protein-U".

- a) a dimerisation factor in making RC-LH1-PufX dimeric;
- b) a "spacer" – to push the several nearby LH1 antenna polypeptides outward from the reaction centre RC.

5) The location of PufX was unambiguously defined within the RC-LH1-PufX monomer of *Rba. sphaeroides* strain IL106.

The PufX is the critical protein required for the photosynthetic growth and participating in dimerisation of *Rhodobacter sphaeroides*.

I consider this work of acute interest for the public.

Major remarks:

1. The evidences of native organisation of the monomer.

Although the structure of the RC-LH1-PufX monomer is perfectly resolved, the question remains about the biological intactness and functionality of the monomer. There is a concern that the monomer described in the paper results from dimer degradation. This degradation potentially could happen during the purification procedure since the dimeric architecture is very fragile.

Questions:

- Could you please add a proof that the monomer is not a result of dimer degradation? Or clarify this possibility in the manuscript?
- Have you observed the RC-LH1-PufX monomer that lacks the PufX during classifications (2D/3D?)
- Have you observed RC-LH1-PufX dimer; LH2 during the 2D/3D classification using cryo-EM technique and/or negative staining technique?

2. The statement that "U-protein" is indispensable for the dimerisation of the RC-LH1-PufX complex.

Although the major claims for the function of U-protein are well explained: dimer stabilisation, "spacer", and RC-LH1-PufX topology controller, the statement that U protein determines dimerisation should be reorganised.

2.1.

As a proof, the mutant of *Rhodobacter* sp. lacking U-protein was purified and analysed.

On the Figure 2, e and on Supplementary Fig. 8a, we still observe the smeared band for RC-LH1_PufX dimer in the sucrose density gradient. It means that the dimers of RC-LH1-PUFX are still present in the U-protein mutant.

Question:

- How would the authors explain that?

On the same gradient figures, we also observe the more significant bend for the monomer.

Question:

- Wouldn't that mean that U-protein serves more for the dimer stabilisation rather than formation?

Suggestion 2.1:

I would suggest restating the statements of the U-protein in the manuscript, stressing more at the stabilisation function in the dimer and not as the reason why the dimer is being formed.

2.2

In the manuscript, pg 12 (249-256), the authors compare the growth of *Rba. sphaeroides* U-protein mutant and *Rba. blasticus* – both those organisms do not have U-protein but can make dimers – for *Rba. sphaeroides* U-protein mutant, we observe the dimer on the sucrose density gradient (Supplementary Fig 8a) and for the *Rba. blasticus* this fact is known from the literature [Scheuring et al., 2005b]

The authors state that the U-protein is not essential for photosynthetic growth (251-252), but the same seems to be true for dimerisation ability.

In the manuscript, the authors showed the presence of U-protein in the monomeric structure of *Rba. sphaeroides* strain IL106.

Question:

- How could you explain the presence of U-protein in the monomeric structure that you revealed?

Suggestion 2.2:

So again, I would suggest to restate the sentences where the function of the U-protein is mentioned. Please see Suggestion 2.1.

3. The Figure containing the numbering of LH1 subunit is required.

Suggestion 3:

In order to enhance the clarity of the manuscript for the reader, please add the Figure containing the numbering of each LH1 antenna subunit. After that, you may update the manuscript accordingly whenever you discuss the interactions, for example, PufX, or U-protein with neighbouring LH1 subunits or between LH1 subunits.

Minor improvements//corrections:

Please, where it is needed, use the name of the studied organism - *Rba. sphaeroides* strain IL106, to avoid misunderstanding of the readers.

43-44 "the complex to block its pores" - please add (channels)

78-79 "the monomer represents a native state in equilibrium with the dimeric complexes" – please add references.

81-82 "*Rba. sphaeroides* LH1 is also unique in its carotenoid composition" – I think you mean "amount"

129 "...opposite each other on the two sides of the LH1-ring opening" – it is good to add the numbering of the LH1 subunits.

129-131 "As a result, the positions of several LH1 $\alpha\beta$ -subunits near protein-U gradually deviated from the closed LH1 ellipse (Fig. 1d)" - the positions of LH1 $\alpha\beta$ subunits are not shown on the Fig. 1d, instead we see the position of Mg atoms.

132-133 "...has been observed in the monomeric type LH1-RCs from *Thermochromatium* (Tch.) *tepidum* and *Rhodospseudomonas* (Rps.) *palustris*. Molecules of BChl a in the *Rba. sphaeroides* LH1 are ligated by His residues with average..." –

Please check the Fig. 1 - there you compare central Mg atom positions in

- *Rba. sphaeroides* (red spheres)
- Tch. *tepidum* (green spheres)
- protein-W-containing Rps. *palustris* (cyan spheres)
- protein-W-deficient Rps. *palustris* (blue spheres)

a) do you mean *Rba. sphaeroides* strain IL106 the one you studying? Please clarify

b) Tch. *tepidum* has a closed LH1 ring. The Rps. *palustris* W - has open ring whereas in Rps. *palustris* without W the ring is closed.

- Why did you placed the Rps. *palustris* without W protein for the comparison since it has also the closed ring, such as Tch. *tepidum*?

139-140 "...modeled in the *Rba. sphaeroides* LH1-RC structure." - do you mean *Rba. sphaeroides*

strain IL106 the one you studying? Please clarify.

152 "...region form hydrogen bonds with two α -polypeptides (Fig. 2c)" – please add the LH1 subunit numbering.

165-166 "...protein-U likely plays a functional role in forming dimers of the LH1-RC complex" – I would suggest to rephrase that sentence.

182-183 "...presumed membrane plane (Fig. 3a, Supplementary Fig. 10)" -- Supplementary Fig. 10a, 10b? Please add.

186-187 "...However, the N-terminal region of PufX interacts with an LH1 α - polypeptide.." – which LH1? Please add numbering.

203-204 "...a similar conformation to those in the LH1 complexes of other purple bacteria (Fig. 4b)" – on the Fig.4b *Tch. tepidum* is shown. Please modify the sentence.

208-209 "a large shift toward the periplasmic side (Fig. 4b)" - Please name in Fig.4b the carotenoid groups according to the manuscript description.

212-213 "The additional Group-B spheroidenes, combined with Group-A spheroidenes.." - What does "additional" Group-B spheroidenes, combined with Group-A means? So there might be "Main one"? Please rephrase this sentence.

227 "The expressed protein-U locates in the interior space between LH1 ..." – please add LH1 subunit numbering.

240-241 "Proteins-U can be classified into three types based on their sequences (Supplementary Fig. 9b)" – why on the Supplementary Fig. 9b the text coloured in 5 different colours? Please change.

266-268 "Rba. sphaeroides strain DBCQG was determined at 8 Å using combined techniques (Quan et al.,2013). However, PufX was found at yet a different position in this structure (Supplementary Fig. 5" – do you mean in comparison with the paper of Quan et al.,2013? Please specify.

293-294 "LH1 α - polypeptide, implying preferential interactions with the LH1 inner ring..." – which LH1 subunit? Please add LH1 subunit numbering.

Figure 1c "Tilted view of the cofactor arrangement..." - Please add from which side do the viewer look and the model, periplasmic?

Figure 4b Please specify here as well which group of carotenoids belongs to Group-A and which to Group-B according to the manuscript's text.

Supplementary Fig. 8 Please add the explanation of the OD = 5; OD = 9; OD = 14 to the figure legend.

Supplementary Fig 9a Please explain the colour code. Why there are 5 different colours?

What does the empty space for the protein U in many columns mean? Does it mean that the protein U does not exist or the protein ID in not known?

Conclusion: Although the argument of the authors that newly found U-protein "is enhancing the dimerization" needs either further clarification or restatement, I recommend this work for publication after the corrections.

Response to reviewers:

Reviewer #1

Reviewer #1's comments

This is a sound piece of structural biology carried out on a monomeric RC-LH1 complex from *Rb. sphaeroides*. The novel features are a better resolution, the presence of a new component 'U' that seems to control the amount of dimer form made, the first visualisation of the two carotenoid molecules per alpha/beta LH1 apoprotein pair and the idea that the second Car stops the possibility of quinone transfer through the LH1 apoprotein barrier. The paper really misses a trick though. It would have been much more significant if a structure had also been determined for the monomer lacking the 'U' component and for the dimer form. Then the really exciting structural role of 'U' and what controls dimer formation could have been seen.

Minor comments;

This is not near atomic resolution as stated in the introduction.

The first ref showing the Car:Bchl_a ratio in the type of complex was Siström W.R. In 'The Photosynthetic Bacteria' pgs. 841-848 (Eds. Clayton and Siström) 1978

Although spheroidene is the major carotenoid when the cells are harvested, membranes are made and the complex is prepared the presence of oxygen results in the conversion of some of the spheroidene into spheroidenone. Did the authors determine the ratio of these two carotenoids to be sure the assumption of just using spheroidene in the structure is correct.

At this resolution it is hard to be sure which type of phospholipid is present. It would be better to reference that the RC has been shown to bind cardiolipin to strengthen this assignment, viz. McAukey et al. (1999) PNAS 96, 14706-14711.

Most readers will not know that the methoxy group at the end of spheroidene is at one end. This should be made much clearer in the text or properly label in a fig.

Our response:

We appreciate the reviewer's positive assessment of our work. Due to the inherent instability of the native dimeric LH1-RC and the LH1-RC lacking protein-U, it has thus far not proven possible to obtain reliable structures for these complexes. Thus, at the moment, we are unable to determine these structures at reasonable resolutions and therefore focused our paper on the monomeric form, where relatively high resolution was obtained.

- We have removed words on the resolution as pointed by the reviewer.
- We have added the reference (Ref. 25) suggested by the reviewer.
- The carotenoid composition in the purified LH1-RC in strain IL106 was the same as that from an LH2-deletion strain DP2 which was previously published in *PNAS* **114**, 10906, 2017 (Ref. 28, Table 1). The spheroidenes were predominant (more than 90%), and spheroidenone was only 3%. Therefore, at the current resolution (2.9 Å) it was reasonable not to model spheroidenones in the structure.
- Since this work was intended to focus on new discoveries (protein-U, PufX and carotenoids), detailed description on the phospholipids was omitted. However, we did mention that 4 cardiolipins and 13 phosphatidylglycerols were identified in our cryo-EM structure of the LH1-RC (p.7). These numbers are in agreement with those determined from biochemical analysis (*BBA* **1860**, 461, 2019). In the revised manuscript, we have added a sentence and a reference as suggested by the reviewer to state that one of the cardiolipins was found at the same position as that reported previously in an RC-only structure from *Rba. sphaeroides* (p.7).
- We have added arrows in Figure 4a to clearly indicate the position of the carotenoid methoxy group, as suggested by the reviewer.

Reviewer #2

Reviewer #2's comments: Point 1

Photosynthesis is the global process by which photosynthetic organisms transform light energy into chemical energy. Photosynthesis is a primary source for the organic synthesis from inorganic matter.

Within various photosynthetic organisms, the purple bacteria place the important niche, as procaryotes, which perform photosynthetic reactions in their way. This feature is one of the reasons why purple bacteria are metabolically the most versatile organisms on Earth.

The interest in the studies of photosynthesis of purple bacteria led to the Nobel Prize in Chemistry 1988 award for Prof. Dr Hartmut Michel, Prof. Dr Johann Deisenhofer and Prof. Dr Robert Huber.

The award was given for the study of the three-dimensional molecular structure and the mode of action of the reaction centre of photosynthesis in the purple bacterium *Rhodospseudomonas viridis*. Ever since the number of research was constantly growing, revealing many new species of purple bacteria and novel organisation of pigment-protein complexes taking part in photosynthetic reactions.

Here I reviewed the manuscript named: "A Novel Membrane Protein in the *Rhodobacter sphaeroides* LH1-RC Photocomplex."

This is a collaborative work performed by an international group of scientists.

The main novelties that are illuminated in the manuscript are:

1) For the first time, the monomeric light-harvesting reaction centre core complex (RC-LH1-PufX) from purple bacteria *Rba. sphaeroides* strain IL106 is revealed at such a high resolution (2.9 Å). Due to such resolution, all the protein secondary structures and many side-chains, pigments and cofactors were undoubtedly defined, modelled and located inside the cryo-EM map (based on the figures//and supplementary figures).

2) A high amount of carotenoids is revealed: each LH1 antenna subunit contains two carotenoids per one α β -polypeptides.

This ratio is the highest of any currently known in the reactions centre of purple bacteria.

3) The structure of unknown density, so-called "protein-U", was found.

Previous attempts to characterise this density were unsuccessful due to its biochemical properties or low resolution of the 3D map. Previous studies defined this density as belonging to the PufX protein.

4) The authors proposed the following explanation of the functioning of "protein-U".

a) a dimerisation factor in making RC-LH1-PufX dimeric;

b) a "spacer" – to push the several nearby LH1 antenna polypeptides outward from the reaction centre RC.

5) The location of PufX was unambiguously defined within the RC-LH1-PufX monomer of *Rba. sphaeroides* strain IL106.

The PufX is the critical protein required for the photosynthetic growth and participating in dimerisation of *Rhodobacter sphaeroides*.

I consider this work of acute interest for the public.

Major remarks:

1. The evidences of native organisation of the monomer.

Although the structure of the RC-LH1-PufX monomer is perfectly resolved, the question remains about the biological intactness and functionality of the monomer. There is a concern that the monomer described in the paper results from dimer degradation. This degradation potentially could happen during the purification procedure since the dimeric architecture is very fragile.

Questions:

- Could you please add a proof that the monomer is not a result of dimer degradation? Or clarify this possibility in the manuscript?
- Have you observed the RC-LH1-PufX monomer that lacks the PufX during classifications (2D/3D?)
- Have you observed RC-LH1-PufX dimer; LH2 during the 2D/3D classification using cryo-EM technique and/or negative staining technique?

Our response:

We thank the reviewer for his/her nice historical review on the bacterial photosynthesis and a brief summary of our work.

- Although there is evidence for the presence of the monomeric LH1-RC in the native *Rba. sphaeroides* membranes (*Nature* **430**, 1058, 2004, Ref. 7), we were unable to isolate and distinguish them from those degraded from dimers, if any, due to the same size and shape. We have therefore added a sentence in the revised manuscript (p.16) to clarify this point as suggested by the reviewer.
- We did not observe the LH1-RC that lacks PufX during both 2D/3D classifications. A slight difference of a single trans-membrane helix was discernible due to the maximum likelihood method used in ordinary 2D/3D classifications. However, as suggested by the reviewer, we have calculated 3D classification in a localized region known as focused classification using the cryo-EM program suite RELION, which could reveal small distinctions between 3D classes (*eLife* **4**, e11182, 2015; *J. Struct. Biol.* **205**, 11, 2019). However, all classes clearly revealed the PufX helix in each density map (See Fig. R1 below). This result illustrates that the LH1-RC monomers always contain PufX.

Fig. R1: Focused 3D classification of the PufX with LH1 pairs. The Class 2 map was determined to 7Å, but other maps had quite low resolutions. However, even at such low resolutions, the density for PufX (orange ribbon) can be easily seen in the maps.

- We did not observe LH1-RC-PufX dimers using cryo-EM/negative staining techniques. However, during 2D classification, we could observe trace amounts of LH2 oligomers and bacterioferritins as contaminants of purification (See Fig. R2 below).

Fig. R2: Typical 2D class averages during iterative 2D classification. Scale bar: 100 Å.

Reviewer #2's comments: Point 2

2. The statement that "U-protein" is indispensable for the dimerisation of the RC-LH1-PufX complex. Although the major claims for the function of U-protein are well explained: dimer stabilisation, "spacer", and RC-LH1-PufX topology controller, the statement that U protein determines dimerisation should be reorganised.

2.1.

As a proof, the mutant of *Rhodobacter* sp. lacking U-protein was purified and analysed.

On the Figure 2, e and on Supplementary Fig. 8a, we still observe the smeared band for RC-LH1_PufX dimer in the sucrose density gradient. It means that the dimers of RC-LH1-PUFEX are still present in the U-protein mutant.

Question:

- How would the authors explain that?

On the same gradient figures, we also observe the more significant bend for the monomer.

Question:

- Wouldn't that mean that U-protein serves more for the dimer stabilisation rather than formation?

Suggestion 2.1:

I would suggest restating the statements of the U-protein in the manuscript, stressing more at the stabilisation function in the dimer and not as the reason why the dimer is being formed.

2.2

In the manuscript, pg 12 (249-256), the authors compare the growth of *Rba. sphaeroides* U-protein mutant and *Rba. blasticus* – both those organisms do not have U-protein but can make dimers – for *Rba. sphaeroides* U-protein mutant, we observe the dimer on the sucrose density gradient (Supplementary Fig 8a) and for the *Rba. blasticus* this fact is known from the literature [Scheuring et al., 2005b]

The authors state that the U-protein is not essential for photosynthetic growth (251-252), but the same seems to be true for dimerisation ability.

In the manuscript, the authors showed the presence of U-protein in the monomeric structure of *Rba. sphaeroides* strain IL106.

Question:

- How could you explain the presence of U-protein in the monomeric structure that you revealed?

Suggestion 2.2:

So again, I would suggest to restate the sentences where the function of the U-protein is mentioned. Please see Suggestion 2.1.

Our response:

- **Response to the questions in 2.1:** There may be a misunderstanding. Deletion of the gene encoding protein-U resulted in a mutant strain that still produces dimeric LH1-RC but at a much-reduced amount (one-third). This can be seen in Fig. 4e and Supplementary Fig. 8a as a denser band for the WT dimers and a much less dense band (probably the "smeared band" referred to by the reviewer) for the dimers of the ΔU -strain as indicated in the figures. Our explanation of the role of protein-U is provisional, including (i) stabilization of the dimer structure as suggested by the reviewer and (ii) enhancement of efficiency for dimer formation. At present, we cannot exclude the possibility that protein-U plays a role in assisting in dimer formation. Therefore, we have modified the statements in the revised manuscript (p.8, 12), as suggested by the reviewer (Suggestion 2.1).
- **Response to the questions in 2.2:** We do not know or wish to speculate why protein-U is present in some *Rhodobacter* species but not in others, and we do not fully understand its accurate functional and structural roles. However, with the structural data we have generated,

we can only predict that protein-U stabilizes (or strengthens) the monomeric LH1-RC by filling the space between LH1 and the RC, and facilitates dimerization of the LH1-RC complex during the expression/assembly processes, likely through subtle interactions with both LH1 polypeptides and the RC L-subunit. To better explain this, we have modified the statements in the revised manuscript (p.8, 12), as suggested by the reviewer (Suggestion 2.2).

Reviewer #2's comments: Point 3

3. The Figure containing the numbering of LH1 subunit is required.

Suggestion 3:

In order to enhance the clarity of the manuscript for the reader, please add the Figure containing the numbering of each LH1 antenna subunit. After that, you may update the manuscript accordingly whenever you discuss the interactions, for example, PufX, or U-protein with neighbouring LH1 subunits or between LH1 subunits.

Our response:

We have replaced Fig. 1b with a new figure that contains numbering (chain ID) for each LH1 polypeptide, protein-U, PufX and the RC LM-subunits. We have also updated the manuscript and figure captions accordingly as suggested by the reviewer.

Reviewer #2's comments: Minor improvements//corrections Part 1

Please, where it is needed, use the name of the studied organism - Rba. sphaeroides strain IL106, to avoid misunderstanding of the readers.

43-44 "the complex to block its pores" - please add (channels)

78-79 "the monomer represents a native state in equilibrium with the dimeric complexes" – please add references.

81-82 "Rba. sphaeroides LH1 is also unique in its carotenoid composition" – I think you mean "amount"

129 "...opposite each other on the two sides of the LH1-ring opening" – it is good to add the numbering of the LH1 subunits.

Our response:

- We have corrected wording and added references according to the reviewer's suggestions.
- We have added numbering in Fig. 1b for each LH1 polypeptide.

Reviewer #2's comments: Minor improvements//corrections Part 2

129-131 "As a result, the positions of several LH1 $\alpha\beta$ -subunits near protein-U gradually deviated from the closed LH1 ellipse (Fig. 1d)" - the positions of LH1 $\alpha\beta$ subunits are not shown on the Fig. 1d, instead we see the position of Mg atoms.

132-133 "...has been observed in the monomeric type LH1-RCs from Thermochromatium (Tch.) tepidum and Rhodospseudomonas (Rps.) palustris. Molecules of BChl a in the Rba. sphaeroides LH1 are ligated by His residues with average..." –

Please check the Fig. 1 - there you compare central Mg atom positions in

- Rba. sphaeroides (red spheres)
- Tch. tepidum (green spheres)
- protein-W-containing Rps. palustris (cyan spheres)
- protein-W-deficient Rps. palustris (blue spheres)

a) do you mean Rba. sphaeroides strain IL106 the one you studying? Please clarify

b) Tch. tepidum has a closed LH1 ring. The Rps. palustris W - has open ring whereas in Rps. palustris without W the ring is closed.

- Why did you place the *Rps. palustris* without W protein for the comparison since it has also the closed ring, such as *Tch. tepidum*?
139-140 "...modeled in the *Rba. sphaeroides* LH1-RC structure." - do you mean *Rba. sphaeroides* strain IL106 the one you studying? Please clarify.

Our response:

- Because the geometry of the LH1 $\alpha\beta$ /BChl *a*-subunit is almost fixed for all LH1 complexes, the positions of central Mg atoms of BChl *a* can be considered to represent the positions of LH1 $\alpha\beta$ -subunits.
- We have made clear that the *Rba. sphaeroides* in Fig. 1d (red) is the strain IL106.
- At the time of submission of this manuscript, the only structure available for a BChl *a*-containing LH1-RC from purple *non-sulfur* bacteria at reasonable resolution was that from *Rps. palustris*, an evolutionarily closer relative to *Rba. sphaeroides* (same α -Proteobacteria). The *Rps. palustris* LH1-RC contains a protein-W instead of the PufX present in *Rba. sphaeroides*. Thus, we placed both W-containing and W-deficient *Rps. palustris* LH1 in Fig. 1d in order to show the structural effect of protein-W on the topology of the LH1 ring. This would in turn imply that our protein-U-deficient LH1-RC of *Rba. sphaeroides* could have a different LH1 structure from what we observed here for the native monomeric complex.
- We have made clear that the *Rba. sphaeroides* strain referred to in this sentence is strain IL106.

Reviewer #2's comments: Minor improvements/corrections Part 3

152 "...region form hydrogen bonds with two α -polypeptides (Fig. 2c)" – please add the LH1 subunit numbering.

165-166 "...protein-U likely plays a functional role in forming dimers of the LH1-RC complex" – I would suggest to rephrase that sentence.

182-183 "...presumed membrane plane (Fig. 3a, Supplementary Fig. 10)" -- Supplementary Fig. 10a, 10b? Please add.

186-187 "...However, the N-terminal region of PufX interacts with an LH1 α -polypeptide.." – which LH1? Please add numbering.

203-204 "...a similar conformation to those in the LH1 complexes of other purple bacteria (Fig. 4b)" – on the Fig.4b *Tch. tepidum* is shown. Please modify the sentence.

208-209 "a large shift toward the periplasmic side (Fig. 4b)" - Please name in Fig.4b the carotenoid groups according to the manuscript description.

212-213 "The additional Group-B spheroidenes, combined with Group-A spheroidenes.." - What does "additional" Group-B spheroidenes, combined with Group-A means? So there might be "Main one"? Please rephrase this sentence.

Our response:

- We have add LH1 subunit numbering and figure numbers, and modified sentences in the text according to the reviewer's suggestion.
- We have rephrased the descriptions of carotenoids (p.9, 10) as pointed out by the reviewer.
- We have added an explanation in the legend to Fig. 4 to indicate the carotenoid groups.

Reviewer #2's comments: Minor improvements/corrections Part 4

227 "The expressed protein-U locates in the interior space between LH1 ..." – please add LH1 subunit numbering.

240-241 "Proteins-U can be classified into three types based on their sequences (Supplementary Fig. 9b)" – why on the Supplementary Fig. 9b the text coloured in 5 different colours? Please change.

266-268 "*Rba. sphaeroides* strain DBC Ω G was determined at 8 Å using combined techniques (Quan et al.,2013). However, PufX was found at yet a different position in this structure (Supplementary Fig. 5" – do you mean in comparison with the paper of Quan et al.,2013? Please specify.

293-294 "LH1 α - polypeptide, implying preferential interactions with the LH1 inner ring..." – which LH1 subunit? Please add LH1 subunit numbering.

Figure 1c "Tilted view of the cofactor arrangement..." - Please add from which side do the viewer look and the model, periplasmic?

Figure 4b Please specify here as well which group of carotenoids belongs to Group-A and which to Group-B according to the manuscript's text.

Supplementary Fig. 8 Please add the explanation of the OD = 5; OD = 9; OD = 14 to the figure legend.

Supplementary Fig 9a Please explain the colour code. Why there are 5 different colours?

What does the empty space for the protein U in many columns mean? Does it mean that the protein U does not exist or the protein ID is not known?

Our response:

- The text colored in 5 different colors was in Supplementary Fig. 9a (not 9b). We have changed the color scheme and provided additional explanation in the empty columns for protein-U in Supplementary Fig. 9a.
- We have added explanation of the OD = 5; OD = 9; OD = 14 to the figure legend in Fig. 8.
- We have added LH1 subunit numbering in the text pointed out by the reviewer.
- We have added language to clarify the viewer's orientation in Fig. 1c.

Additional edits

- The title has been changed to avoid use of "novel".
- A new reference (Ref. 24) and corresponding statement (p. 9) have been added on a recently published LH1-RC-PufX structure from *Rba. veldkampii*, a close relative to *Rba. sphaeroides*. *Rba. veldkampii* does not contain protein-U (Supplementary Fig. 9).
- Supplementary Fig. 5 has been modified to incorporate structure comparison of PufX between *Rba. sphaeroides* strain IL106 (our work) and *Rba. veldkampii*.

REVIEWER COMMENTS

Reviewer #2 (Remarks to the Author):

The authors of the reviewed article "A Previously Unrecognized Membrane Protein in the Rhodobacter sphaeroides LH1-RC Photocomplex" have performed considerable revisions that improved the structure and the description of accomplished research.

Few remarks to consider –

115 – mainchain – Please change to main chain

191 - C-terminal domain has closer contacts – probably contact.

367 - a manner that exclude the protein-U gene and cloned – excludes/or excluded, please correct.

393 - on a glow-discharged holey carbon – article "a" is excessive. Consider removing it.

I am convinced that the reviewed article is complete and ready for submission in Nature Communication.